# Morphological Changes of Glial Lamina Cribrosa of Rats Suffering from Chronic High Intraocular Pressure

**DOI:** 10.3390/bioengineering9120741

**Published:** 2022-11-30

**Authors:** Jingxi Zhang, Yushu Liu, Liu Liu, Lin Li, Xiuqing Qian

**Affiliations:** 1School of Biomedical Engineering, Capital Medical University, Beijing 100069, China; 2Beijing Key Laboratory of Fundamental Research on Biomechanics in Clinical Application, Capital Medical University, Beijing 100069, China

**Keywords:** high intraocular pressure, glial lamina cribrosa, pore area fraction, morphology

## Abstract

Deformations or remodeling of the lamina cribrosa (LC) induced by elevated intraocular pressure (IOP) are associated with optic nerve injury. The quantitative analysis of the morphology changes of the LC will provide the basis for the study of the pathogenesis of glaucoma. After the chronic high-IOP rat model was induced by cauterizing episcleral veins with 5-Fluorouracil subconjunctival injection, the optic nerve head (ONH) cross sections were immunohistochemically stained at 2 w, 4 w, 8 w, and 12 w. Then the sections were imaged by a confocal microscope, and six morphological parameters of the ONH were calculated after the images were processed using Matlab. The results showed that the morphology of the ONH changed with the duration of chronic high IOP. The glial LC pore area fraction, the ratio of glial LC pore area to the glial LC tissue area, first decreased at 2 w and 4 w and then increased to the same level as the control group at 8 w and continued to increase until 12 w. The number and density of nuclei increased significantly at 8 w in the glial LC region. The results might mean the fraction of glial LC beam increased and astrocytes proliferated at the early stage of high IOP. Combined with the images of the ONH, the results showed the glial LC was damaged with the duration of chronic elevated IOP.

## 1. Introduction

Glaucoma, the primary cause of irreversible blindness worldwide, is a progressive optic neuropathy with cupping of the optic nerve head (ONH), thinning of the retinal nerve fiber layer, and loss of the visual field. It is predicted to affect more than 1.1 million people in 2040 [1]. With the progression of the disease, loss of the retinal ganglion cell (RGC) axons, morphological changes of the ONH, and distortion of lamina cribrosa (LC) are discovered [2,3]. Although the pathogenesis of glaucoma is still unclear, pathological intraocular pressure (IOP) elevation is considered one of the risk factors for glaucoma. The elevated IOP may lead to ON damage because it will induce LC deformation or remodeling, compressing the optic nerve (ON). Therefore, the study of morphological changes of the LC with high IOP will lay a fundament for the pathogenesis of glaucoma.

The lamina cribrosa, which is regarded as the primary damage site for glaucoma, is a sieve-like tissue composed of collagenous fibers in the ONH [4]. The unmyelinated portion of the optic nerve extends into the LC and becomes the myelinated ON after passing through the LC. Fortified astrocytes and RGC axons are the main components of the rat glial LC area [5]; they play a mechanical load-bearing role in protecting and supporting the RGC axons for the rat eyes. Studies have found that the thickness of the LC decreased and the anterior surface depth of the LC increased, which indicated that the LC may move backward and deform with an increased IOP [6,7,8]. In addition, structural alternations of the LC might impact axoplasmic transport or blood flow, leading to irreversible damage [9,10,11].

The clinical appearance of the LC in glaucomatous eyes may continue to change based on the clinical examination in vivo. A retrospective study showed that differences in LC pore shape and size of glaucomatous eyes have also been associated with damage to ON [12] using optic disc photographs. Based on images obtained by swept-source optical coherence tomography (OCT), Omodaka et al. found that glaucoma patients had a smaller area and 3D volume of LC pores, probably due to the expanding area of connective tissue [13]. Elongation of lamina pores was more evident in primary open-angle glaucoma eyes than in normal eyes [14,15].

Acute IOP rise may induce a change in LC morphology. Previous studies proposed that the microstructure of the LC, including collagen bundles and pores, was altered after a short-term rise in IOP, in which collagen fiber orientation changed and a lateral movement of LC pores occurred [16,17]. A study using computational modeling showed that as IOP increased, the size of LC pores increased and the shape of pores turned more convex [18]. Ling et al. [5] found a significant change in the area fraction of glial fibrillary acidic protein (GFAP), which could label astrocytes, in the unmyelinated region three days after IOP elevation. The increase in connective tissue area may be related to the proliferation of astrocytes, which constitute the main glial structure of the LC [19].

The development of glaucoma is progressive, so it is important to study the chronic elevation of IOP to simulate the pathogenesis of glaucoma. Reynaud et al. [20] observed that the LC pore diameter and connective tissue volume increased in monkey early experimental glaucoma eyes. LC tissue damage and changed morphology were found in a rodent chronic elevated IOP model [21,22]. However, how LC microstructure changed under chronic high IOP is still unclear.

The arrangement of collagen fiber bundles at the LC is not uniform. Ling et al. [23] showed that the peripheral region of the LC had a higher pore area fraction than the central part, and strain (except for radial–circumferential shear strain) increased with an increase in pore area fraction, which indicated that the LC beam density was related to IOP-induced strain. Similarly, localized tissue damage on the dorsal region of the rat LC was observed at the early stage of damage under the high IOP [22]. In addition to regional distinctions, some studies found that the elongation index of LC pores and the radial displacement on the anterior surface of the LC were significantly higher, indicating inhomogeneous stress on LC [14,15]. Ling et al. [5] measured the structural features of unmyelinated ON region variations with axial locations in the mouse optic nerve after short-term IOP elevation.

Therefore, the axial variation rules of the LC microstructure under chronic high IOP need to be further investigated to comprehensively understand the microstructural changes of the LC region, which can provide a basis for early damage for glaucoma. The fortified astrocytes, which are the main component of the rat LC tissue, are called “glial LC” [24,25]. This study aims to better understand the microstructure or morphology changes of the glial LC along the axial locations under sustained IOP elevation by measuring morphological parameters, including ONH area, pore area fraction, aspect ratio, and the number of nuclei.

## 2. Materials and Methods

### 2.1. Animals

Thirty adult male SD rats, weighing about 300 g, were acquired from the Animal Department of Capital Medical University and kept in an environment with suitable temperature and humidity and standard water and food. All rats were randomly divided into 5 groups, namely a control group and four experimental groups (2 w, 4 w, 8 w, and 12 w after chronic high-IOP model induction). They were examined for intact corneas, normal intraocular pressure, and no other ocular diseases before the experiment. All experiments followed the Regulations for the Administration of Affairs Concerning Experimental Animals issued by the State Scientific and Technological Commission of China, and all animals were treated in strict accordance with the ARVO Statement for the Use of Animals Ophthalmic and Vision Research.

### 2.2. Model Induction and IOP Measurement

All IOP measurements were conducted in awake rats using a Tonolab tonometer (iCare, Vantaa, Finland) at a fixed time (9 a.m.–10 a.m.). The probe was kept perpendicular to the cornea during the measurement. A total of three measurements were recorded and the average value was taken as the final IOP. Baseline IOP was measured before induction.

The modeling method was described in detail in previous studies [26,27]. The rats were anesthetized intraperitoneally with 1% Pelltobarbitalum Natricum prepared with saline (40 mg/kg), and the cornea was anesthetized with Oxybuprocaine Hydrochloride Eye Drops (Santen, Osaka, Japan). Then we cauterized episcleral veins of the right eyes until the distal end of the vessels became white due to ischemia and the proximal end of the vessels became congested, while the left eyes were considered as contralateral control eyes. We performed 5-Fluorouracil (Shanghai Xudong Haipu Pharmaceutical Co., Ltd., Shanghai, China) subconjunctival injection to inhibit the regeneration of neovascularization after cauterization. IOP was measured every three days after induction. If IOP was lower than 30 mmHg, cauterization was performed again. Otherwise, only 5-Fluorouracil was injected.

### 2.3. Staining and Imaging

ONH was sectioned, stained, and then captured in the control group and experimental groups, including 2 w, 4 w, 8 w, and 12 w after model induction. After intraperitoneal injection with an overdose of anesthetic, eyeballs were quickly removed. A 4 × 4 mm tissue where the sclera intersected the optic nerve with 1–2 mm optic nerve was preserved in phosphate-buffered saline (PBS). The tissue was fixed in 4% paraformaldehyde for 12 h and then in 30% sucrose for 12 h. During both the fixation and dehydration steps, the tissue was stored in a 4 °C refrigerator.

After being dried, the tissue was embedded in OCT (Sakura Finetek USA, Torrance, CA, USA) compound along the ON and then frozen in liquid nitrogen. ONH was sectioned continuously into 10 μm sections from the Bruch’s membrane opening (BMO) at −20 °C using a freezing microtome (Leica, Wetzlar, Germany), and there were 16 sections for one sample.

After being removed from the refrigerator, cryosections were placed on a baking machine at 37 °C for 30 min to prevent desquamation and then washed three times for 5 min in PBS. A circle around the tissue was drawn with a pap pen to avoid edge effects. Cryosections were blocked in 5% bovine serum albumin (BSA; Sigma, Saint Louis, MO, USA) with 0.3% Triton X-100 in PBS for 1 h at room temperature and transferred to primary rabbit anti-GFAP antibody solution which was diluted with 5% BSA in PBS (1:500, AB7260, Abcam, Cambridge, UK). The sections were incubated in the primary antibody solution overnight at 4 °C in a humidity chamber. After primary incubation, the sections were rewarmed for 30 min at room temperature and then washed four times for 5 min in PBS. Then the sections were incubated in goat secondary anti-rabbit antibody solution with 1% BSA (1:500, AB150077, Abcam, Cambridge, UK) for 2 h at 37 °C. After the application of the antifade mounting medium with DAPI (Abcam, Cambridge, UK), the staining sections were finalized.

Under a ×63 oil objective, fluorescent images were captured at 4096 × 4096 pixels by a confocal microscope (Leica, Wetzlar, Germany) in three channels, as shown in Figure 1.

### 2.4. Image Processing

A series of image processing was completed using Matlab (MathWorks, Natick, MA, USA). A 3 × 3 median filter was applied to all images to degrade isolated noise and smooth the images by replacing the intensity of every pixel with the median intensity of surrounding pixels. For GFAP channel images, we applied the Jerman filter, which is based on the Hessian matrix and Gaussian kernel, to enhance the contrast of the glial LC beam structure (Figure 2g) [28,29]. To enhance the nuclei channel images, we used contrast-limited adaptive histogram equalization (CLAHE) following the median filter. All images were binarized to segment the glial LC beams and nucleus from the black background by Otsu’s method which can convert a grayscale image to a binary image by calculating minimum interclass variance as the threshold between the foreground and background [30] (Figure 2c,h). After binarization, a series of morphological processes were applied to remove or fill isolated pixels and bridge unconnected pixels to make them fit the real image. Finally, we used a dilation followed by an erosion to blend narrow gaps or bends and then fill gaps in the profile (Figure 2d, yellow arrows in Figure 2i). Considering the cell adhesion, we performed a watershed transform method to find ridge lines between two closing cells to separate contiguous regions into distinct objects, which can improve the accuracy of counting nuclei (yellow arrows in Figure 2e) [31].

### 2.5. Structural Measurements

In this study, a 160 μm thick area from the unmyelinated region along the optic nerve was selected. Six morphological features [5] of the ONH were calculated after the images were processed using Matlab:

ONH Area: The total area within the ONH boundary which was detected from the processed GFAP images by tracing the outer boundary of the filled region (Figure 3a).

GFAP Area: The area labeled by GFAP in processed GFAP channel images within the ONH boundary.

Pore Area Fraction: The ratio calculated as the pore area, which was equal to the ONH area minus the GFAP area, over the ONH area.

Aspect Ratio: The ratio of the length to width of the minimum external rectangle of the ONH. The minimum external rectangle of the ONH is shown in Figure 3b as red.

Number of Nuclei: The number of separate regions in the ONH in the nuclei channel labeled by DAPI.

Nuclear Density: The number of nuclei divided by 10^3^ per square millimeter in the ONH area.

### 2.6. Regional Division

The ONH was sectioned from the Bruch’s membrane opening (BMO) and was divided into two regions: the pre-laminar region and the glial LC region [32]. According to the aspect ratio results (Figure 4a), the pre-laminar region was defined by a distance of 0 to 40 μm from the BMO, and the morphology of the ONH was closest to the circle type (Figure 4b,c). The other region was defined as the glial LC region, which was defined by a distance of 40 to 160 μm from the BMO (Figure 4d–g).

### 2.7. Statistical Analysis

Data were expressed as mean ± standard deviation. IOP was averaged in 6 rats in each group and compared with contralateral control eyes using paired t-test. One-way ANOVA was used to compare the differences in six structural features between the control group and the other 4 groups of chronic high IOP. Bonferroni corrections were performed after one-way ANOVA multiple comparisons. All analyses were performed using SPSS 26.0 (IBM Corporation, Armonk, NY, USA) and a *p* value lower than 0.05 was considered statistically significant.

## 3. Results

IOPs of both eyes were measured when the rats were awake before the experiment. Normal IOPs were about 12.50 ± 0.29 mmHg, and the values were similar in both eyes (*p* > 0.05). IOPs of experimental eyes increased after the chronic high-IOP model was induced, and the high IOP level was sustained until 12 w. The contralateral control eyes maintained normal IOP, indicating that the method of inducing the model did not affect the IOP level of the contralateral eyes in this study (Figure 5).

As shown in fluorescent images of cross sections in the glial LC region, the glial LC was kidney-shaped, with the inward depression as the ventral surface and the other side as the dorsal surface (Figure 6a). The processed images (Figure 6f–j) showed the variation in the glial LC pore area fraction with the chronic elevated IOP. The enlarged images showed the variation in the dorsal surface (Figure 6k–o) and the ventral surface (Figure 6p–t). The astrocytes were arranged in an orderly radial pattern under normal IOP, and pores were clear (Figure 6a,k,p). At 2 w and 4 w after IOP elevation, fibers became distorted (red arrows) and thicker (red asterisks). Disorganized fibers (white arrows) and blurred GFAP staining (yellow arrows) appeared, along with abnormal tissue gaps due to fiber loss (white asterisks) under the elevation of IOP in experimental groups. Meanwhile, the radioactive form of the glial LC beam was completely lost, indicating that the arrangement of astrocytes had been disrupted by sustained high IOP. It was shown that the GFAP distribution of the dorsal surface was disordered with high IOP, indicating that the tissue damage was more pronounced in the dorsal region as well as at the end of the glial LC than in the ventral region.

The pore area fraction decreased gradually along the axial direction of ON, and it was significantly higher in the pre-laminar region than in the region behind at 4 w (*p* < 0.01) and 8 w (*p* < 0.05, Figure 7a). The ONH area gradually increased after 120 μm from BMO except at 2 w and 4 w (Figure 7b). Similarly, the GFAP area also increased after 120 μm from BMO except at 4 w (Figure 7c). In addition, there was no clear trend of change in the number and density of nuclei along the ON (Figure 7d,e).

In this study, all sections along the ON were separated into two regions based on the aspect ratio, namely the pre-laminar and glial LC regions. Then all features were averaged from sections in the glial LC region. Pore area fraction within the glial LC region decreased at 2 w and 4 w after model induction (*p* < 0.01) and then increased at 12 w (*p* < 0.01), compared to the control group (Figure 8a). The processed images (Figure 6f–j) could confirm that the fibers seemed denser at 2 w and 4 w, while disorganized fibers and blurred GFAP staining along with abnormal tissue gaps appeared at 12 w, which might cause the increase in glial LC pore area fraction. As for the contralateral control eyes, there was no significant difference between groups (*p* > 0.05, Figure 8a). Compared to the contralateral control eyes, the glial LC pore area fraction of experimental eyes decreased at 4 w and then increased at 12 w significantly (*p* < 0.01, Figure 8b). Two 50 × 50 μm areas were selected from similar positions in both ventral and dorsal regions of glial LC (red and white boxes in Figure 8c) in each section. The glial LC pore area fraction of the selected region was then averaged in the glial LC region of six rats in each group. It was found that there was no significant difference in glial LC pore area fraction between the dorsal surface and the ventral surface (*p* > 0.05). The glial LC pore area fraction decreased at 2 w and 4 w and then increased until 12 w in both dorsal and ventral regions with continuous high IOP (Figure 8d).

The ONH area, GFAP area, number of nuclei, and density of nuclei within the glial LC region, as previously defined by a distance of 40 to 160 μm from the BMO, were counted and calculated automatically. The name of the ONH region within the glial LC region was simplified as the glial LC tissue region. These features within the glial LC region from six rats in each group were averaged (Table 1). The number of nuclei in the glial LC region increased at 8 w of high IOP (*p* > 0.05) and then decreased significantly (*p* < 0.05, vs. 8 w experimental group). The nucleus density increased with the duration of high IOP, reaching a maximum at 8 w (*p* < 0.05), and then decreased at 12 w (*p* < 0.05, vs. 8 w experimental group). The glial LC tissue area decreased until 4 w (*p* > 0.05) and then increased at 8 w (*p* > 0.05). There was no significant difference in GFAP area between groups (*p* > 0.05). Compared to contralateral control eyes, there was no significant difference in the glial LC tissue area in four experimental groups (*p* > 0.05, Figure 9a). However, the density of nuclei in the glial LC region of experimental eyes increased at 8 w (*p* < 0.01) and then declined at 12 w (*p* < 0.01) compared to contralateral control eyes (Figure 9b).

## 4. Discussion and Conclusions

This study presented some information on glial LC variations with the duration of high IOP by extracting morphological parameters based on the immunohistochemical staining of frozen cross sections of the ONH. We obtained the morphological parameters of glial LC along the axial direction by image processing and also determined the variation in morphological parameters of glial LC at different time periods under chronic high IOP.

The high IOP level of the model in this study could be sustained for three months, and we selected four time periods (2 w, 4 w, 8 w, and 12 w) to observe the morphology of the glial LC of the model. Many forms of glaucoma have no warning signs. Glaucomatous optic nerve damage is ongoing but remains asymptomatic until later stages. If glaucoma is recognized early, vision loss can be slowed or prevented. The previous studies [33,34] showed that the LC was the primary damage site for glaucoma. Scanning electron microscopic analysis suggests that the structure of the lamina cribrosa is an important determinant of the degree of susceptibility to damage by elevated intraocular pressure [34]. The changes in the morphology of the LC under the effect of chronic elevated IOP could be researched using rodent hypertensive glaucoma models [21]. By injection of microbeads, laser photocoagulation, or episcleral vein cauterization, a chronic high intraocular pressure animal model can be induced [11,31,32,35]. We induced the model by cauterizing episcleral veins with 5-Fluorouracil subconjunctival injection, which might elevate the episcleral venous pressure and then obstruct the outflow of aqueous humor. This model, widely used for the study of glaucoma, is reproducible and simple to manipulate with fewer complications [21,36]. We hope to obtain the typical feature of morphological changes of animal LC at the different sustained times of high IOP to map the course of glaucoma. The elevated IOP could be sustained from 10d to 24 w [11,21,31,32,35]. Considering that the change in LC morphology occurred the early stage of injury, we have studied the axonal transport of the optic nerve using a confocal laser scanning microscope and the morphology of the optic nerve head of a rat after 4 w of high IOP using hematoxylin–eosin (HE) staining, immunofluorescence staining, and transmission electron microscopy (TEM) [12]. The results showed that the activated astrocytes might squeeze the optic nerve to lead to optic nerve distortion and axonal flow blockage. An increased optic nerve injury grade positively correlated with increased microglia/macrophage density in anterior and transition ONH was found after 5 w of chronic high IOP [37,38]. In order to measure the change in LC morphology in detail, we chose four time periods (2 w, 4 w, 8 w, 12 w) to research in this study.

In this study, we performed continuous transverse sectioning of a 160 μm thick region close to the BMO within the unmyelinated region and then observed the clear microstructure of the glial LC using a confocal microscope. Several imaging methods can be used in vivo to study the morphology or structure of the LC [4,39], including confocal scanning laser ophthalmoscopy [40], ultrasound [15], and optical coherence tomography (OCT) with several modalities such as spectral-domain OCT, swept-source OCT, and enhanced depth imaging OCT [41,42,43]. Since the measurement resolution of LC pores using OCT in vivo is often lower than that of using a confocal microscope in vitro, many investigations [5,32] prefer to apply a confocal microscope. Further, [5,44] showed that the 200 μm thick region near the BMO in the rat ONH was found to be an unmyelinated region with strong GFAP immunoreactivity. However, the thickness of rodent glial LC and pre-laminar region in the unmyelinated area is still inconclusive [19,37]. A previous study regarded the region where the shape of glial LC was circular as the “pre-laminar” region [32]. In order to separate the pre-laminar region from the ONH and quantify the glial LC morphology features, we fit the minimum external rectangle of it for aspect ratio calculation, which is applicable for regular or irregular figures in many fields [45,46]. The results of this study showed that the aspect ratio of the ONH in all groups increased with the distance from the BMO and that the shape of the LC was similar to the kidney type. Since the aspect ratio of the ONH tended to be 1 with circular morphology within the region 0–40 μm from the BMO, this region was regarded as the “pre-lamina” region in this study.

The results of this study showed that there was no significant difference in the glial LC pore area fraction between the dorsal surface and the ventral surface in the glial LC region. Ling et al. [23] discovered that the porosity and LC pore fraction of the periphery were higher than those of the interior region in the human LC by using second harmonic generation imaging. Although in this study there was no significant difference in the pore area fraction between the two regions, it could be confirmed by fluorescence images that the tissue on the dorsal region was damaged initially with severe tissue defect, which is similar to the previous study [22]. In addition, the glial LC pore area fraction in the pre-laminar region was significantly higher than that in the glial LC region at 4 w and 8 w. This might be due to the greater pressure on the anterior glial LC which stretched and deformed glial LC pores [14]. The ONH area and GFAP area gradually increased after 120 μm from BMO except at 4 w. However, a significant increase in ONH area was found in a previous study [5]. Under normal intraocular pressure, the collagen bundles demonstrated a radial pattern. With the continued effect of high intraocular pressure, this regular morphological feature was disrupted, and the overall morphology of glial LC changed from renal to irregular, demonstrating that the elevation of IOP could cause redirection of astrocytes. In addition, Tehrani et al. found that the redirection of astrocyte protrusions in response to early elevated IOP was both axon-dependent and -independent [37,38].

Under elevated intraocular pressure, the glial LC pore area fraction first decreased before 4 w, indicating that the size of glial LC pores became smaller compared with that in the control eyes at the early stage of high IOP. Meanwhile, the network structure of astrocytes was significantly remodeled, and the glial LC skeleton was denser. This might be a response to pressure, as it was previously noted that regions with a high density of collagen beams exhibited lower strain [23]. In addition, it might be due to the increased expression of GFAP. A study claimed that the increase in GFAP expression and hypertrophy of astrocytes in moderate reactive astrogliosis might lead to the impression of an increase in astrocytes [47]. Morrison et al. [48] have found that astrocytes filled in the atrophied region of the optic nerve axons and phagocytose dead neurons with the continuous development of glaucoma in an animal model, and then astrocytes directly synthesized the extracellular matrix, while the optic nerve fibers disappeared. Elevated IOP will lead to increased ONH strain [49,50], reactive phenotype, and activation of astrocytes, including hypertrophy, proliferation, migration, and release of extracellular matrix [51,52,53,54], and then more astrocytes are produced with a denser meshwork of glial LC to reduce deformation. Reynaud et al. [20] found that connective tissue volume and LC volume were larger in monkey early experimental glaucoma. However, a study showed that the size of LC pores increased and the shape of pores turned more convex under increased IOP [18]. The results of this study also showed the feasibility of this assertion. Under the further action of pressure in this study, the glial LC pore area fraction rose, indicating that the pore space became larger and glial LC tissue was severely damaged. We speculated that the increase in the glial LC pore area fraction might be due to nerve damage or reactive astrocytes which might cause the variation in the morphology of the glial LC. Quigley et al. [34] discovered a decrease in the number of axons and loss of fibers in glaucoma by light and transmission electron microscopy. The previous studies [22,55] showed that the number of RGC axons decreased significantly after chronic high IOP by cross-section staining. In severe nerve injury, there is the formation of a glial scar associated with tissue reorganization and structural changes [47]. In addition, abnormal tissue gaps and blurred GFAP staining at 12 w were evident from the fluorescence images, leading to the increase in glial LC pore area fraction.

In this study, significant increases in the number and density of nuclei in the glial LC region were found at 8 w, indicating cell proliferation at the late stage of injury. A previous study found that there was hypertrophy of a single cell body without astrocyte proliferation in mild or moderate reactive astrogliosis. As reactive astrogliosis became severe, there was astrocyte proliferation along with compact glial scar formation [47]. Because of the non-specificity of DAPI staining, we could not figure out the exact type of proliferating cells. It has been confirmed that the astrocyte is the predominant cell type in the ONH, along with microglia and macrophages [56]. Astrocytes serve as the primary and earliest force sensors of ONH injury in glaucoma; their response to injury precedes other symptoms of glaucoma and may ultimately contribute to axonal degeneration in glaucoma [57]. A study of cells in the ONH of the glaucoma model through specific staining has shown that the RGC axons are lost and the number of nuclei in the anterior ONH increases for the advanced injury ONH, even reaching twice that of the control group. Moreover, the proportion of microglia and macrophages increased, while the proportion of astrocytes decreased in the anterior ONH as the injury progresses [19], which may explain the finding in this study that the number of nuclei increased but glial LC pore area fraction first decreased and then increased due to changes in the number of astrocytes. Bosco et al. also found that the microgliosis in the ONH was in the late stage of severe injury under the DBA/2J model [40].

In summary, we measured the changes in glial LC structure along the axial direction with the duration of high IOP in this study. After the chronic high-IOP rat model was induced, the cross sections of the ONH along the axial direction were immunohistochemically stained at 2 w, 4 w, 8 w, and 12 w. Six morphological parameters of the ONH were calculated after the images were processed using Matlab. The results showed the variation in the glial LC pore area fraction was not monotonous, which was consistent with the changes in nucleus number and density in the glial LC region. Compared to the contralateral control eyes, the glial LC pore area fraction of experimental eyes decreased at 4 w and then increased at 12 w significantly. Meanwhile, the density ratio of nuclei increased with the duration of high IOP, reaching a maximum at 8 w and then decreasing at 12 w. There are some limitations in this study. Astrocytes are the primary cell type in the ONH of the rat eye, but the effects of other cells on the glial LC, such as microglia and LC cells, were ignored in this study because of the nonspecific staining. It had been proved that LC cells could be labeled by staining actin to improve the accuracy analysis of ONH cells [58]. In addition, there were no morphological features of glial LC pores being measured, which might also change with IOP elevation [5,18]. In this paper, we analyzed and evaluated the changes in glial LC morphology based on two-dimensional images. If we can achieve three-dimensional reconstruction of the ONH in different periods of high-IOP effects, more comprehensive results may be obtained.

## Figures and Tables

**Figure 1 bioengineering-09-00741-f001:**
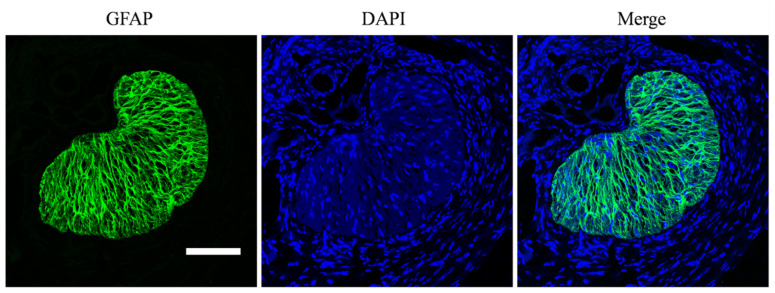
Fluorescent images were captured using a confocal microscope in three channels, namely the GFAP channel in which the glial LC beams were stained green, the DAPI channel in which the cell nucleus was blue, and the merged image. Scale bar: 100 μm.

**Figure 2 bioengineering-09-00741-f002:**
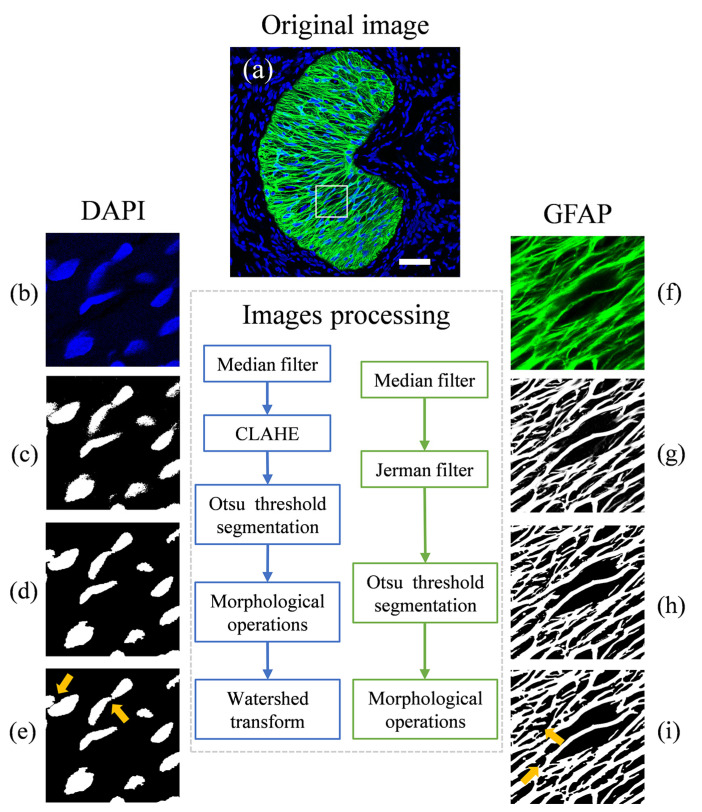
The images in both GFAP and DAPI channels were processed to measure the morphological features of the ONH. (**a**) Original image, with a 50 × 50 μm area marked as a white box to show the specific results. The image processing flow was put below (**a**), and the blue box shows the processing flow for the DAPI channel, while the green one is for the GFAP channel. (**b**,**f**) are enlarged images (white box in (**a**)) in DAPI and GFAP channels, respectively. For GFAP channel images, the median filter and Jerman filter were applied (**g**). Then they were binarized by Otsu’s method to segment the glial LC beams (**h**). Finally, various morphological operations were applied to remove isolated pixels and bridge unconnected pixels (yellow arrows in (**i**)). The white pixels represent glial LC beams, and the black pixels represent glial LC pores. For the nucleus channel images, a median filter along with CLAHE and Otsu’s method was used to smooth, enhance, and segment it (**c**). Then a set of morphological operations were performed (**d**). Lastly, we applied a watershed transform method to segment overlapping cells (yellow arrows in (**e**)). The separated white pixel regions represent a nucleus. Scale bar: 50 μm.

**Figure 3 bioengineering-09-00741-f003:**
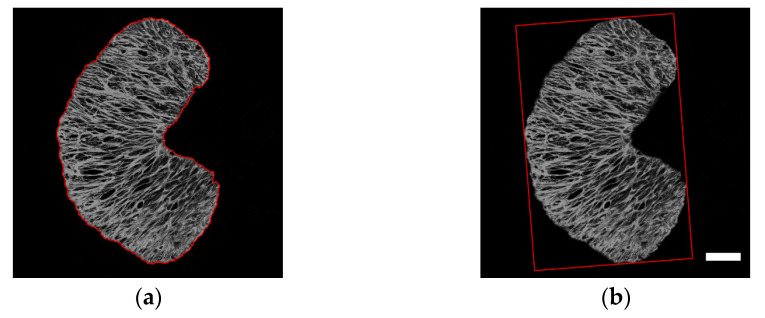
The boundary (**a**) and the minimum external rectangle (**b**) of the ONH were marked red automatically on the grayscale image. Scale bar: 50 μm.

**Figure 4 bioengineering-09-00741-f004:**
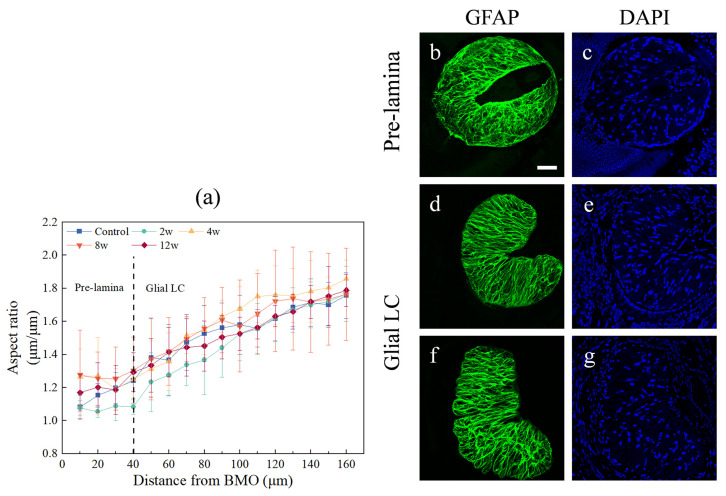
The aspect ratio and two-channel images along the ON. (**a**) The aspect ratio of the ONH gradually increased along the ON. All 16 sections of each rat along the ON were separated into two groups, namely the pre-laminar region (0 to 40 μm from BMO) and glial LC region (40 to 160 μm from BMO); (**b**,**c**) 10 μm from BMO; (**d**,**e**) 50 μm from BMO; (**f**,**g**) 160 μm from BMO. Scale bar: 50 μm.

**Figure 5 bioengineering-09-00741-f005:**
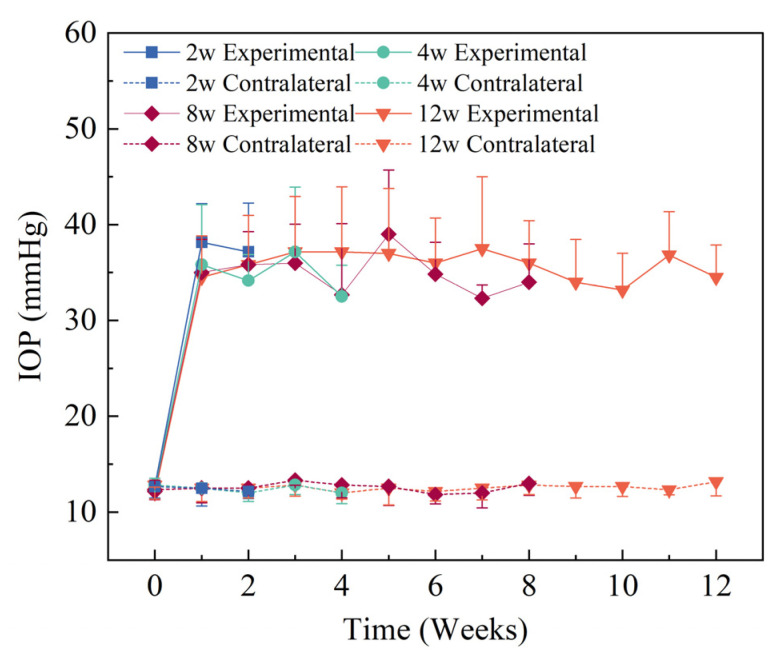
IOPs of both eyes in the experimental groups (*n* = 6). The IOPs of experimental eyes increased significantly after model induction (*p* < 0.05) and the IOPs of contralateral control eyes remained at a normal level (*p* > 0.05).

**Figure 6 bioengineering-09-00741-f006:**
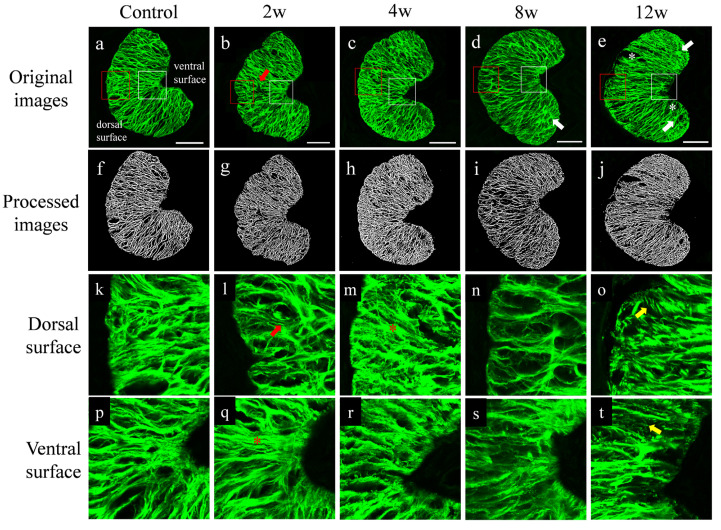
The fluorescent images and processed images of cross sections in all groups. (**a**–**e**) Overall images in control and experimental groups, showing the variation in the meshwork of astrocytes (green). (**f**–**j**) The processed images in all groups, indicating the change in glial LC pore area fraction under high IOP. The 80 μm × 80 μm areas in the dorsal surface (**k**–**o**) from the red boxed areas in (**a**–**e**) and the ventral surface (**p**–**t**) from the white boxed areas. The astrocytes were arranged in an orderly radial pattern with clear pores under normal IOP. With increasing IOP, distorted fibers appeared (red arrows) and fibers became thicker (red asterisks). There were disorganized fibers (white arrows) and blurred GFAP staining (yellow arrows), along with abnormal tissue gaps due to fiber loss (white asterisks). Scale bar: 80 μm.

**Figure 7 bioengineering-09-00741-f007:**
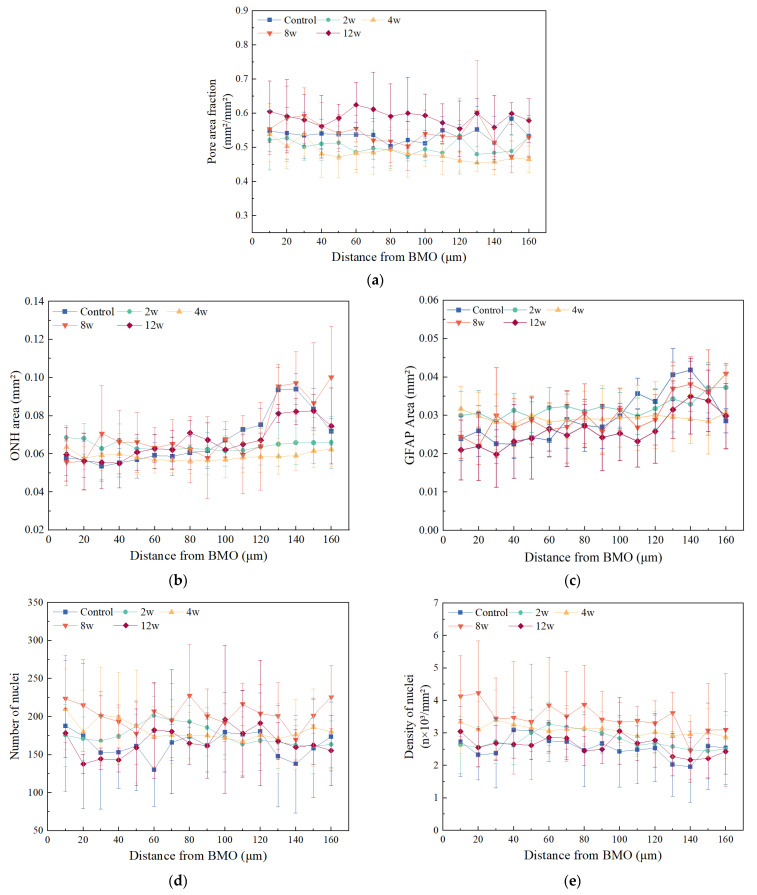
The variation in the morphological features along the ON is shown, including (**a**) pore area fraction, (**b**) ONH area, (**c**) GFAP area, (**d**) number of nuclei, and (**e**) density of nuclei. The features at each position were obtained by averaging the experimental eye values of six rats in each group.

**Figure 8 bioengineering-09-00741-f008:**
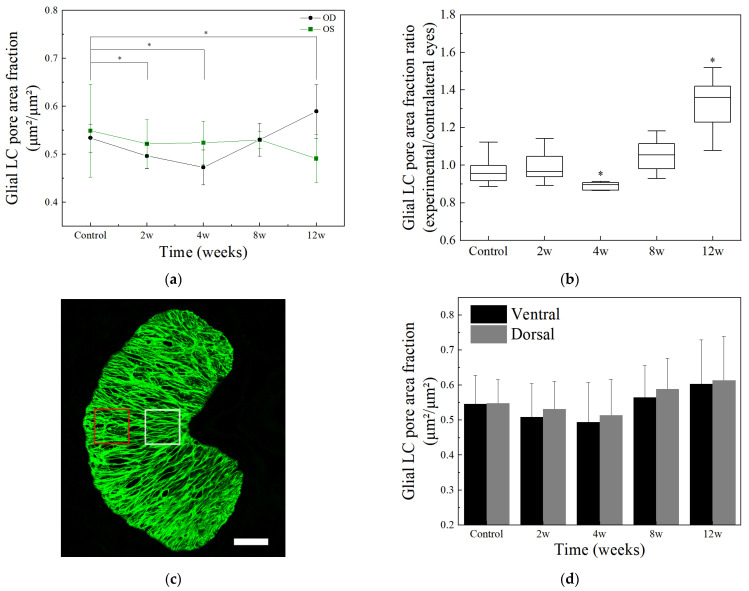
The pore area fraction of the glial LC, averaged from 6 rats in each group, changed with the effect of chronic high IOP. (**a**) The pore area fraction in the glial LC region decreased at 2 w and 4 w and then increased under high IOP (*p* < 0.01). The glial LC pore area fraction of the contralateral control eyes had no significant difference between groups (*p* > 0.05). (**b**) The glial LC pore area fraction ratio of experimental eyes to contralateral control eyes, an average of sections in six rats per group. (**c**) A fluorescent image in the GFAP channel with a selected area in the ventral region (white box) and the dorsal region (red box). (**d**) The variation in the glial LC pore area fraction in two regions, ventral and dorsal surface, with an elevation of IOP and comparison between two regions (*p* > 0.05). Scale bar: 50 μm. * *p* < 0.01 indicates statistical significance.

**Figure 9 bioengineering-09-00741-f009:**
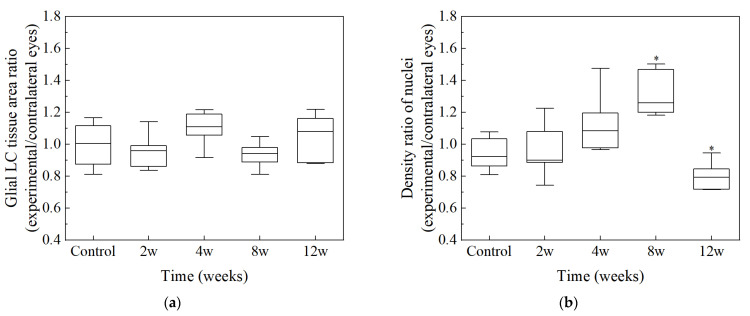
Comparison of the glial LC tissue area (**a**) and density of nuclei (**b**) between experimental eyes and contralateral control eyes in the glial LC region. * *p* < 0.01 indicates statistical significance.

**Table 1 bioengineering-09-00741-t001:** The glial LC tissue area, GFAP area, and number and density of nuclei in glial LC region (*n* = 6).

Groups	Number of Nuclei	Glial LC Tissue Area (mm^2^)	GFAP Area (mm^2^)	Density of Nuclei (*n* × 10^3^/mm^2^)
Control	164 ± 21 ^ab^	0.0660 ± 0.00911	0.0271 ± 0.00487	2.67 ± 0.392 ^b^
2 w	176 ± 18 ^ab^	0.0648 ± 0.00679	0.0320 ± 0.00495	2.76 ± 0.241 ^ab^
4 w	169 ± 28 ^ab^	0.0595 ± 0.00613	0.0296 ± 0.00657	3.00 ± 0.403 ^ab^
8 w	201 ± 7 ^a^	0.0703 ± 0.00811	0.0313 ± 0.00414	3.43 ± 0.467 ^a^
12 w	162 ± 17 ^b^	0.0612 ± 0.00521	0.0234 ± 0.00710	2.57 ± 0.300 ^b^

Bonferroni corrections were performed after multiple comparisons; numbers marked by any same letter (for instance, ‘a’ and ‘a’, ‘b’ and ‘b’, ‘ab’ and ‘ab’, ‘a’ and ‘ab’, ‘b’ and ‘ab’) indicate a non-significant difference between the two groups (*p* > 0.05), while different letters (for instance, ‘a’ and ‘b’) mean a significant difference (*p* < 0.05). The absence of letters means there is no significant difference between all groups.

## Data Availability

No new data were created or analyzed in this study. Data sharing is not applicable to this article.

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
