# Peer review of "Morphological Changes of Glial Lamina Cribrosa of Rats Suffering from Chronic High Intraocular Pressure"

_bioengineering, 2022, doi:10.3390/bioengineering9120741_

Round 1

Reviewer 1 Report

The research article by Dr Zhang, Dr Liu et al., entitled “Morphological changes of glial lamina cribrosa of rat suffered 2 from chronic high intraocular pressure”, deals with the alterations of the optical nerve head and lamina cribrosa induced by elevated intraocular pressure (IOP) in rats following cauterization of episcleral veins. The study is well designed and experiments are organized adequately. Results obtained support the conclusions.

I have only several comments concerning the opportunity to implement data presentation.

1)      It would be of interest to graphically represent how the pore area fraction changes following the increase of the IOP.

2)      It seems that the number and the density of the nuclei do not change significantly during increasing time intervals, while the glial pore area increases during the time. How the Authors explain these results?

3)      Apart from glial population, is there any evidence of the occurrence of a lesion of nerve fibers? The Authors should discuss about the potential presence of a reduction of nerve fibers in their model and whether this may correlate with the increase of the glial pore area.

4)      In the methods section, the Authors state that polyformaldeyde was used to fix the tissues dissected from rats. I did not hear before about this compound. Please, could they specify more about this?

Reviewer 2 Report

In this manuscript, the authors investigated the morphology of glial LC through imaging in a chronic high IOP rat model. The analysis included the number of nuclei, the area of glial LC tissue, the area of GFAP, and the density of nuclei. Despite the fact that the study contains a great deal of valuable information, monotonous image analysis appears to be limited in its ability to provide in-depth information. There is a lack of understanding regarding the physiological and pathological implications of the four time periods (2w, 4w, 8w, 12w) analyses described above. What was the reason for analyzing only three months? The results of week 8 show a significant change, but why not weeks 2, 4, and 12? At week eight, a change should be explained in terms of its physiological significance. The Glial LC pore area fraction ratio is the lowest at 4 weeks and the highest at 12 weeks, but it is necessary to explain why. Also, in the comparison between ventral and dorsal (fig.8d), the 4th and 12th weeks are not significant. What is the reason for this?

Reviewer 3 Report

In this study, the authors investigated the remodeling of lamina cribrosa induced by chronic elevated intraocular pressure in rats. The cross sections of optic nerve head were immunohistochemically stained at 2w, 4w, 8w, and 12w, along with a control group. The authors concluded that the glial LC were damaged by the duration of chronic elevated IOP. There are some major concerns that need to be addressed before it could be further processed.

Major points:

1.     The study claimed that deformation was observed in glial LC, as measured with multiple morphological features. However, the elevated IOP may also increase the LC depth (LCD), i.e., the position of LC measured based on the BMO or peripapillary sclera, which has been shown on human. The accuracy of arbitrary division of pre-lamina and glial LC (40μm from BMO) may be influenced after the LCD change. A longitudinal section data may help to clarify such issues. 

2.     Multiple comparison among groups should be corrected by post-hoc analysis or Bonferroni correction after the ANOVA.

Minor points

Methods:

1.     In Figure 2, the processing details of GFAP can be emphasized in 2f-2i by arrows as in 2e.

2.     In 2.6, why did the authors define the 40μm from BMO as the cut-off between pre lamina and glial LC. Please clarify as reference 32 did not provide the relevant details.

Results:

Overall, please provide P value in detail if mentioned in the article.

1.     Line 217, please provide detail IOPs in mean±SD of the baseline.

2.     Figure5: Is there any immediate IOP data after the model induction?

3.     Figure6: The descriptions in line 226-236 were not that apparent in Figure 6. Details can be emphasized with asterisks or arrows in the figure.

4.     Line 242-247: How did the trend of change in axial direction analyze? Please clarify the statistical methods used here. Please provide the P values of ONH area, GFAP area. Better to label in the figure. 

5.     Line 257: Please clarify how to select position of the 50*50 μm block within the dorsal or ventral region. Were they in the same position among all the samples or randomly chosen?

6.     Line 260-262: Please rephrase the sentence. It was ambiguous whether there was significant difference.

7.     Figure8: 8a: The glial LC pore area fraction of contralateral eyes should be added. 8b: Add the data of the control group. 8d & line262-263: Add P value. Plus, the P<0.05 can be changed to P<0.01 in the figure. 

8.     Table 1: P values vs experimental group can be marked in other symbols and emphasized in the table.

9.     Figure 9: Add the data of the control group

Discussion and conclusion

1.     The authors mentioned a trend of variation along the axial direction of ONH. Further descriptions can be provided for such trend.

2.     Line 357: Why did the authors define 8w as a late stage of injury?

3.     The conclusion is not precisely organized. Besides, the novelty can be addressed more clearly.

Misspelling should be checked. For instance, the “scar bar” in Figure 4, 6 & 8 should be changed to “scale bar”.

Round 2

Reviewer 2 Report

Thanks to the authors for their revisions. I believe that it is suitable for publication.

Reviewer 3 Report

The revision is acceptable.